# Analyzing Transfer Characteristics of Disordered Polymer Field-Effect Transistors for Intrinsic Device Parameter Extraction

## Minho Yoon

Department of Physics and Institute of Quantum Convergence Technology, Kangwon National University, Chuncheon 24341, Republic of Korea; minhoyoon78@gmail.com

**Abstract:** In this study, we present an intrinsic device parameter method based on a single device for disordered polymer field-effect transistors (PFETs). Charges in disordered polymer semiconductors transport through localized states via thermally activated hopping, of which field-effect mobility and contact resistance are gate-bias-dependent. By considering the parameters expressed as gate bias-dependent power laws, dividing drain current with transconductance ($I_{ds}/g_m$ method) leads to the current–voltage relation decoupled from the contact effect. Following this derived relationship, the intrinsic field-effect mobility and the contact resistance of the PFETs are extracted and found to be consistent with those using the four-probe method. Thus, we can state that the proposed method offers practical benefits for extracting the intrinsic device parameters of disordered PFETs in terms of using a single transfer characteristic of the devices.

**Keywords:** polymer field-effect transistors; charge transport; device parameters





## 1. Introduction

In recent decades, solution-processable polymer field-effect transistors (PFETs) have received considerable attention due to their massive potential for applications in flexible displays, logic circuits, and wearable devices [1–4]. Especially, donor–acceptor (D–A)-type semiconducting copolymers containing donor or acceptor moieties such as diketopyrrolopyrrole (DPP), isoindigo, naphthalenediimides (NDI), benzothiadiazole, and indacenodithiophene (IDT), exhibit remarkable field-effect mobilities exceeding $10 \ cm^2 \ V^{-1} \ s^{-1}$ [5–9]. However, the inevitable contact resistance in transistors generally results in under or overestimating field-effect mobility [10–12]. Thus, the device parameters should be carefully determined. To extract device parameters decoupled from the contact effects, several approaches have been suggested, such as measuring potential distribution in the channel by placing the voltage probes (i.e., four-probe method) [13], extracting the channel conductance by varying the channel length (i.e., transmission line measurement or transfer length measurement) [14], and eliminating contact effects using the drain current and transconductance (i.e., $Y$-function method) [15]. While there is progress, they often require specific channel geometries, rendering their general application to PFETs. In the case of the differential method [16,17], TFTs that have different channel lengths should be used for the extraction. However, if we use multiple devices for the parameter extraction, the deduced results can be erroneous due to uniformity issues. In addition, the $Y$-function method could be used for the extraction using a single device, but the results could be carefully examined for disordered semiconductors [18]. The $Y$-function method is established for Si-based transistors, of which the parameters are considered to be gate-bias independent. However, due to the localized states, the parameters of the disordered semiconductor-based PFETs are gate-bias dependent [19,20]. For these reasons, we regard simple and reliable intrinsic parameter extraction methods using a single device are required for disordered semiconductor-based PFETs.

Here, we explore an intrinsic device parameter method for disordered polymer field-effect transistors (PFETs). By considering the field-effect mobility and contact resistance as

the gate bias-dependent power-laws [21,22], the current-voltage relation decoupled from the contact effect is derived by dividing the drain current with transconductance ($I_{ds}/g_m$ method). With this relation, the contact resistance and intrinsic mobility are successfully extracted using a single device, and the parameters are found to be consistent with those by the four-probe method. Hence, we regard our proposed method as a simple and reliable extraction method using a single device for disordered PFETs.

## 2. Materials and Methods

Bottom-gate, top-contact (BGTC) polymer field-effect transistors (PFETs) were fabricated, as shown in Figure 1a. A chemically cleaned 300 nm-thick $p^+$-Si/SiO$_2$ wafer was used as the substrate, and hexamethyldisilazane (HMDS) was spin-casted and thermally annealed at 150 °C for 60 min in a nitrogen atmosphere to minimize the interface traps [23]. Poly [2,5-(2-octyldodecyl)-3,6-diketopyrrolopyrrole-alt-5,5-(2,5-di(thien-2-yl) thieno [3,2-b]thiophene)], (DPP-DTT, Mw (~75,000) with a PDI of 2.5), and poly(2,5-bis(3-tetradecylthiophen-2-yl)thieno [3,2-b]thiophene) (PBTTT, Mw (~50,000) with a PDI of 3.0) were purchased from 1-Material, Inc and Sigma-Aldrich, respectively. Then, polymer semiconducting materials were dissolved in chloroform at concentrations of 10 mg/mL, spin-casted, and subsequently thermally annealed at 150 °C for 60 min in nitrogen. Next, 50 nm thick Au source and drain electrodes were deposited by thermal evaporation and patterned using a shadow mask. For a passivation layer, poly (methyl methacrylate) PMMA was dissolved in n-butyl acetate at a concentration of 80 mg/mL, spin-casted, and cured at 80 °C for 60 min. Finally, the layers were patterned by reactive etching with Ar gas to mitigate the side-current effects on the devices [24]. The channel width and length of PFETs were 1000 and 360 µm, respectively. The surface morphologies of the films were scanned with an atomic force microscope (XE100, Park Systems, Suwon, Korea). The geometric capacitance of the dielectric was measured to be 10.9 nF cm$^{-2}$ at 1 kHz using an LCR meter (HP4284A, Agilent Technologies, Santa Clara, CA, USA). The current-voltage (*I–V*) characteristics of the transistors were investigated with a semiconductor parameter analyzer (Model HP4155C, Agilent Technologies). The extrinsic field-effect mobility of the PFETs in the linear regime was extracted using Equation (1).

$$\mu_{lin} = \frac{1}{C_i V_{ds}} \frac{L}{W} \frac{\partial I_{ds}}{\partial V_{gs}} \tag{1}$$

where $I_{ds}$ is the drain current, $V_{gs}$ is the gate voltage, $C_i$ is the geometric dielectric capacitance, $V_{ds}$ is the drain voltage, and $L$ and $W$ are the channel length and width, respectively. In addition, by measuring the potentials in the channel with the voltage probes ($V_1$ and $V_2$), the intrinsic field-effect mobility ($\mu_{\text{int}}$) and contact resistance ($R_c$) were estimated, which are given by Equations (2) and (3) [13,18]:

$$V_{ch} = \frac{(V_2 - V_1)}{(L_2 - L_1)}(L), \qquad V_c = V_{ds} - V_{ch} \tag{2}$$

$$\mu_{int} = \frac{1}{C_i V_{ch}} \frac{L}{W} \frac{\partial I_{ds}}{\partial V_{gs}}, \qquad R_c = \frac{V_c}{I_{ds}} \tag{3}$$

where $V_{ds}$, $V_1$, and $V_2$ are the drain voltage and measured voltages with potential probes, and $L_1$, $L_2$, and $L$ are the distance from the source electrode to the first, second, and drain electrodes, respectively. In this study, the voltage probing electrodes ($V_1$ and $V_2$) were placed at 120 and 240 µm in the channel.

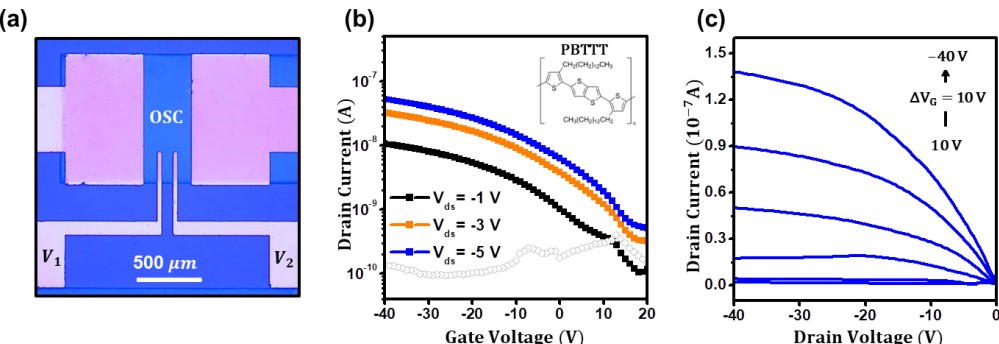

**Figure 1.** (**a**) Optical microscopy image of the polymer field-effect transistors (PFETs) with a four-point probe configuration. (**b**) Transfer characteristics ($I_{ds}$ vs. $V_{gs}$) of PFETs. Inset: chemical structures of the PBTTT. (**c**) Output characteristics ($I_{ds}$ vs. $V_{gs}$) of the PBTTT-based PFETs.

## 3. Results

Figure 1b displays the representative transfer characteristics ($I_{ds}$ vs. $V_{gs}$) of the PBTTT-based PFETs; it exhibits p-type characteristics, of which the field-effect mobility ($\mu_{ext}$) from the transconductance was estimated to be ~0.009 cm$^2$ V$^{-1}$ s$^{-1}$ along with a high on/off ratio (>10$^3$). The gate leakage current level was maintained as low as 10$^{-10}$ A. Figure 1c presents the corresponding output characteristics ($I_{ds}$ vs. $V_{ds}$) of the PFETs. Then, for extracting the intrinsic referential parameters of the PFETs, we investigated the transfer characteristics with the four-probe method. Please see the material and methods section for more information on the four-probe method. Figure 2a shows the measured channel and contact potentials as a function of the gate bias at the drain bias of –1, −3, and −5 V. As seen in Figure 2b,c, regardless of the applied drain bias, the contact resistance ($R_c$) was gradually decreased to be ~8.9 × 10$^5$ $\Omega$ cm at $V_{gs}$= −40 V, and the intrinsic field-effect mobility ($\mu_{int}$) was estimated to be ~0.010 cm$^2$ V$^{-1}$. Thus, it is highly believed that the gate-bias-dependent intrinsic device parameters are precisely extracted by the four-probe method. However, placing the voltage–probing electrodes in specifically designed electrode geometries, such as interdigitated or circle-type source-drain electrodes, can be challenging. In addition, to our best knowledge, device parameter extraction methods using a single device for disordered semiconductors are scarcely reported [16]. Hence, we have tried to establish reliable extracting methods for disordered semiconductor-based thin-film transistors using a single transfer characteristic.

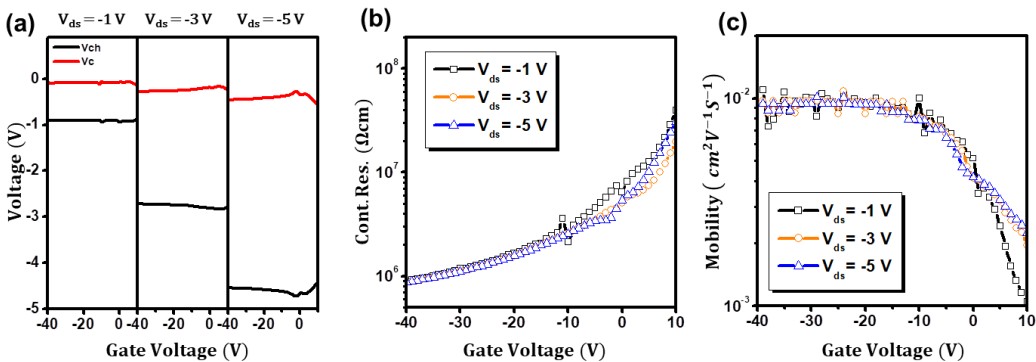

**Figure 2.** (**a**) Measured potential distributions of the PFETs with a four-probe configuration. (**b**,**c**) Extracted contact resistance and mobility of the PFETs from the four-probe method, respectively.

As reported elsewhere, charges in disordered semiconductors transport through localized states via thermally activated hopping, of which the intrinsic field-effect mobility ($\mu_{int}$) and the contact resistance ($R_c$) can be expressed as gate bias-dependent power laws in Equation (4) [25–28]. In addition, as depicted in Figure 2a, the voltage drop ($I_{ds}R_c$) is considered almost independent of the gate bias, which leads to the condition $\alpha + \beta + 1 = 0$.

Hence, the drain current ($I_{ds}$) and transconductance ($g_m$) can be expressed as gate bias-dependent power laws as in Equations (5) and (6):

$$\mu_{int} = \mu_{int,0}\left(V_{gs} - V_{th}\right)^{\alpha}, \qquad R_c = R_0\left(V_{gs} - V_{th}\right)^{-(\beta)} \tag{4}$$

$$I_{ds} = \frac{\mu_{int,0}C_i\frac{W}{L}\left(V_{gs} - V_{th}\right)^{\alpha+1}V_{ds}}{1 + \mu_{int,0}R_0C_i\frac{W}{L}} \tag{5}$$

$$g_m = \frac{\partial I_{ds}}{\partial V_{gs}} = \frac{(\alpha + 1)\mu_{int,0}C_i\frac{W}{L}\left(V_{gs} - V_{th}\right)^{\alpha}V_{ds}}{1 + \mu_{int,0}R_0C_i\frac{W}{L}} \tag{6}$$

where $\mu_{int,0}$, $R_0$, and $\alpha$ are the coefficients and the exponents of the power law of the intrinsic field-effect mobility and the contact resistance, respectively. Then, the coefficient of $\alpha$ and the threshold voltage ($V_{th}$) can be estimated by dividing the drain current with the transconductance ($I_{ds}/g_m$) as in Equation (7),

$$\frac{I_{ds}}{g_m} = \frac{\left(V_{gs} - V_{th}\right)}{(\alpha + 1)} \tag{7}$$

Moreover, the coefficient of contact resistance ($R_0$) can be given by Equations (8) and (9), of which the drain current ($I_{ds}$) can be described in terms of extrinsic and intrinsic mobility.

$$I_{ds} = \frac{\mu_{int,0}C_i\frac{W}{L}\left(V_{gs} - V_{th}\right)^{\alpha+1}V_d}{1 + \mu_{int,0}R_0C_i\frac{W}{L}} = \mu_{ext}C_i\frac{W}{L}\left(V_{gs} - V_{th}\right)V_{ds} \tag{8}$$

$$\begin{aligned} R_0 &= \left(\frac{(1+\alpha)}{\mu_{ext,0}} - \frac{1}{\mu_{int,0}}\right)\frac{L}{C_iW} \quad for\ V_{gs} - V_{th} = 1\ V \\ &\approx \left(\frac{(\alpha)}{\mu_{ext,0}}\right)\frac{L}{C_iW} \quad if\ \mu_{int,0} \approx \mu_{ext,0} \end{aligned} \tag{9}$$

where $\mu_{ext,0}$ is the coefficient of the extrinsic field-effect mobility. After deducing the coefficient contact resistance ($R_0$), the contact-effect removed drain current ($I_{ds,int}$) can be determined by removing the contact resistance from the total resistance, and the resulting intrinsic mobility can be extracted as in Equations (10) and (11), respectively.

$$I_{ds,\ int} = \frac{V_{ds}}{R_{ch}} = \frac{V_{ds}}{\frac{V_{ds}}{I_{ds}} - R_0\left(V_{gs} - V_t\right)^{-(\alpha+1)}} \tag{10}$$

$$\mu_{int} = \frac{1}{C_iV_D}\frac{L}{W}\frac{\partial I_{ds,int}}{\partial V_{gs}} \tag{11}$$

Figure 3a shows the extraction plot of $I_{ds}/g_m$ for PBTTT-based PFETs. The coefficient of $\alpha$ and the threshold voltage ($V_{th}$) is extracted to be 0.51 and 9.41 V, 0.56 and 11.33 V, and 0.59 and 11.82 V, at the drain biases of −1, −3, and −5 V, respectively. In addition, the coefficient contact resistance ($R_0$) was extracted to be 0.85, 2.55, and $2.30 \times 10^9\ \Omega$ cm at the drain biases, respectively. Then, the contact resistances and intrinsic mobilities were estimated as in Figure 3b, 3c, and 3d, respectively. Although the deduced coefficients of $\alpha$ and the threshold voltages ($V_{th}$) are slightly different at the given biases, the consequential contact resistances ($R_c$) and the intrinsic field-effect mobilities ($\mu_{int}$) are deduced to be almost the same values; $R_c$ was gradually decreased to be ~$3.2 \times 10^6\ \Omega$ cm at $V_{gs}= -40$ V, and $\mu_{int}$ was extracted to be ~0.015 cm$^2$ V$^{-1}$, which comparable with those obtained by the four-probe method. Please note that when we tried to extract parameters using the *Y-function* method for the PFETs, as depicted in Figure S1, only gate-bias-independent parameters were estimated, of which the field-effect mobility was slightly overestimated to be 0.017 cm$^2$ V$^{-1}$, and the contact resistance was underestimated to be $2.1 \times 10^6\ \Omega$ cm, respectively. As estimated by the 4-probe method and the $I_{ds}/g_m$ method, the parameters should be gate-dependent. Thus, we strongly believe that the proposed extraction method

of the $I_{ds}/g_m$ method is simple and promising for extracting the intrinsic parameters using a single device. Please note that the intrinsic mobility from the 4-probe method and the extrinsic mobility of the HMDS-treated PBTTT TFTs (at $V_{gs} - V_{th}$ = 1 V) was almost the same as extracted to be 0.0012 and 0.0013 cm$^2$ V$^{-1}$ s$^{-1}$, respectively. Thus, we tried to derive the relations based on the assumption in Equation (9) ($\mu_{int,0} \approx \mu_{ext,0}$). However, it is still unconfirmed whether the proposed extraction method is reliable for other PFETs. Hence, to ensure that the $I_{ds}/g_m$ method is applicable in more general, we have fabricated PFETs using another organic semiconductor of DPP-DTT, and the device parameters of the PFETs are analyzed in the same manner.

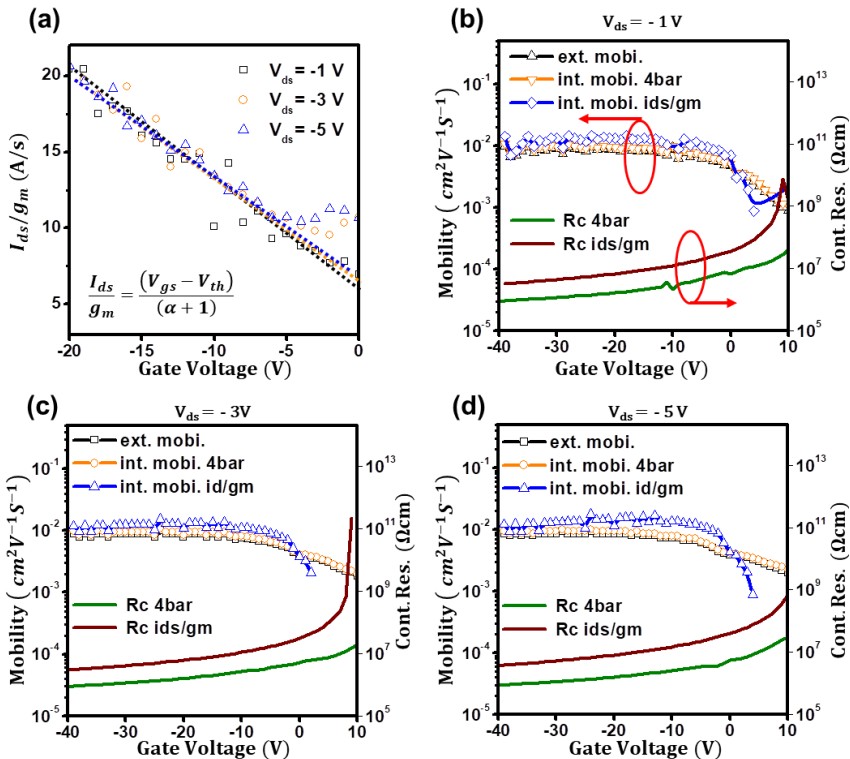

**Figure 3.** (**a**) The extraction plot for PBTTT-based PFETs. (**b**–**d**) Extracted contact resistance and mobility of the PFETs from the $I_{ds}/g_m$ method at the drain biases of −1, −3, and −5 V, respectively.

Figure 4a shows the representative transfer characteristics of the DPP-DTT PFETs. It displays p-type characteristics, and the extrinsic field-effect mobility is estimated to be ~0.021 cm$^2$ V$^{-1}$ s$^{-1}$ with a high on/off ratio (>10$^4$). Figure 4c displays the extraction plot of $I_{ds}/g_m$ for DPP-DTT PFETs, of which the coefficient of $\alpha$ and the threshold voltage ($V_{th}$) is extracted to be 0.68 and −5.91 V, respectively. Remarkably, although the DPP-DTT PFETs show Schottky characteristics in output characteristics of Figure 4b, intrinsic mobility, and contact resistance are estimated to be almost the same as ~0.029 cm$^2$ V$^{-1}$ and ~1.5 × 10$^6$ Ω cm at $V_{gs}$= −40 V with those by the 4-probe method in Figure 4d. This indicates the proposed method can effectively rule out the contact effect from the drain current. Furthermore, to gain insights into the device analyses using the method, we have fabricated octadecyltrichlorosilane (OTS)-treated DPP-DTT PFETs, and the device parameters are compared with those of the HMDS-treated DPP-DTT PFETs. Figure 5a shows the representative transfer characteristics of the PFETs. Figure 5b,c exhibit the estimated intrinsic motilies and contact resistances of the PFETs by the $I_{ds}/g_m$ method, respectively. Noteworthy, compared to HMDS-treated PFETs, OTS-treated PFETs exhibit enhanced p-type characteristics. The hole current at $V_{gs}$ = −40 V increases over ten times (from ~10$^{-7}$ to ~10$^{-6}$ A), and the threshold voltage of the PFETs clearly shifts from −5.91 to 1.35 V. As a result; the field-effect mobility massively improves up to 0.351 cm$^2$ V$^{-1}$ s$^{-1}$. In addition, the mobility of the OTS-treated PFETs increases in the given applied bias regime,

which can be expressed with the power law. However, the field-effect mobility of the HMDS-treated PFETs increases up to the gate bias of −15 V, but slightly decreases beyond the voltage, probably due to charge scattering in the channel [29–31]. We posit that these enhancements of the PFETs would be mainly attributed to the improved coplanarity and interchain connectivity of the donor–acceptor (D–A)-type conjugated copolymers by the surface treatment with OTS. As seen in Figure 5c,d, nano-fibril microstructures with large grains can definitely be found in OTS-treated DPP-DTT films. The root mean square (RMS) surface roughness of OTS-treated DPP-DTT films was measured to be 0.87 nm, while that of HMDS-treated DPP-DTT films was 0.62 nm. Hydrophobic characteristics of the OTS, as in Figure S2, would allow the conjugated copolymers to interact and aggregate strongly with each other, leading to enhanced intermolecular interactions. As a result, the charge scattering is minimized in the PFETs, enhancing the field-effect mobility. Moreover, the contact resistance of the OTS-treated PFETs was also significantly decreased to ~$10^5$ Ω cm, which compared to that of the HMDS-treated PFETs of ~$10^6$ Ω cm. The enhanced ordering of the polymers leads to improved charge carrier injection from the source electrode to the π-electron orbitals of the polymers, decreasing the contact resistance of the PFETs [32,33]. The extraction plot and the output characteristics ($I_{ds}$ vs. $V_{ds}$) of the OTS-treated DPP-DTT PFETs are depicted in Figure S3. The observed device performances of the PFETs appear primitive, and parameter estimations using other polymer semiconductors, such as n-type or ambipolar semiconducting polymers, would be required to prove the method is applicable in more general. In addition, probably due to the charge scattering, the deduced contact resistance and mobility by the $I_{ds}/g_m$ method are slightly under and overestimated than those by the 4-probe method. As the gate bias increases, the discrepancy between the observed mobility and the power-law modeled field-effect mobility ($\mu_{int} = \mu_{int,0}(V_{gs} - V_{th})^{\alpha}$) would increase and result in the underestimation of the coefficient of α. As a result, the deduced contact resistance and mobility could be slightly under and overestimated than those by the 4-probe method. For PBTTT PFETs, the channel scattering effect could be greater than those in D–A type semiconducting copolymers of DPP-DTT, resulting in more deviated parameters. However, the estimated contact resistances by both methods are in order, and the differences in the mobilities are in the suborder of $10^{-3}$ cm$^2$ V$^{-1}$ s$^{-1}$. Thus, we regard the parameter extraction method of the $I_{ds}/g_m$ method as reliable. The extracted device parameters, such as field-effect mobility and contact resistance of the devices, are summarized in Table 1.

**Table 1.** Summary of device parameters for the PFETs used in this study.

| Semiconductors | Extraction Methods | $\mu$ (cm$^2$ V$^{-1}$) | Power Law Exponent | $R_c$ at −40 V (Ω cm) |
|---|---|---|---|---|
| HMDS-treated PBTTT | Trans. method | 0.009 | Not available | Not available |
| | Four-probe method | 0.010 | Not available | $8.9 \times 10^5$ |
| | $I_{ds}/g_m$ method | 0.015 | 0.59 | $3.2 \times 10^6$ |
| | Y-function method | 0.017 | Not available | $2.1 \times 10^6$ |
| HMDS-treated DPP-DTT | Trans. method | 0.023 | Not available | Not available |
| | Four-probe method | 0.029 | Not available | $1.6 \times 10^6$ |
| | $I_{ds}/g_m$ method | 0.028 | 0. 68 | $1.2 \times 10^6$ |
| OTS-treated DPP-DTT | $I_{ds}/g_m$ method | 0.351 | 0.18 | $4.8 \times 10^4$ |

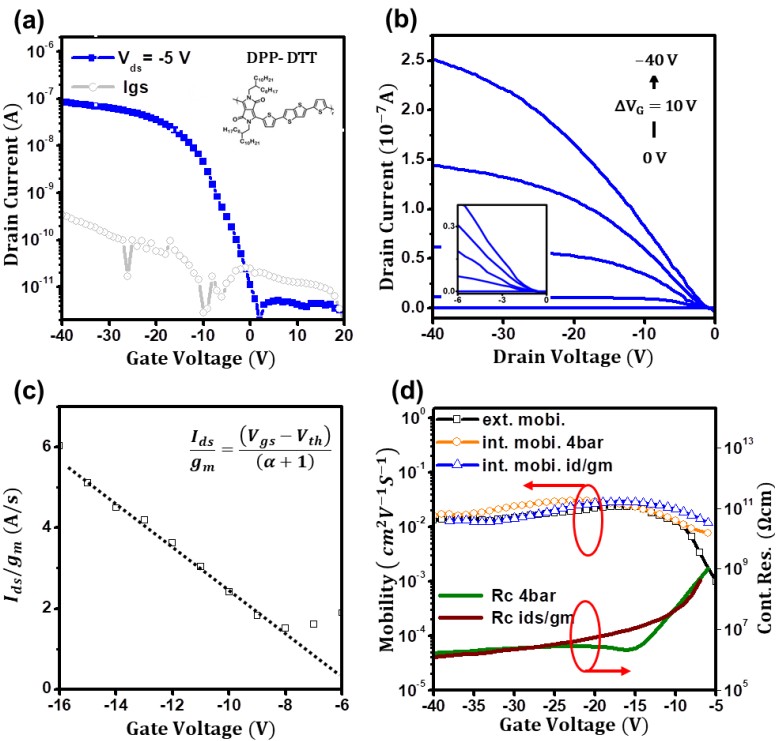

**Figure 4.** (**a**) Transfer characteristics ($I_{ds}$ vs. $V_{gs}$) of the DPP-DTT PFETs. (**b**) Output characteristics of the DPP-DTT PFETs. Inset: Output characteristics in a low drain bias regime. (**c**) The extraction plot of $I_{ds}/g_m$ for the PFETs. (**d**) Extracted contact resistance and mobility of the PFETs from the $I_{ds}/g_m$ method.

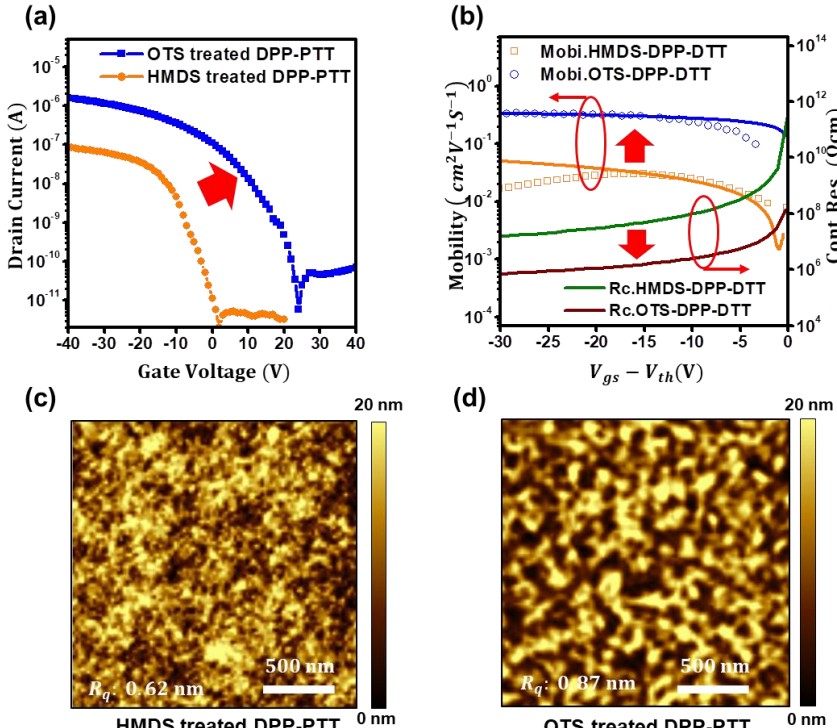

**Figure 5.** (**a**) Transfer characteristics ($I_{ds}$ vs. $V_{gs}$) of the OTS-treated DPP-DTT PFETs. (**b**) Extracted mobility and contact resistance of the OTS-treated PFETs from the $I_{ds}/g_m$ method. (**c**,**d**) AFM images of HMDS and OTS-treated DPP-DTT films, respectively.

## 4. Conclusions

In this study, we investigate an intrinsic device parameter extraction method for disordered polymer field-effect transistors (PFETs). Due to the localized states of the disordered polymer semiconductors, the field-effect mobility and contact resistance are strongly dependent on the gate bias. In addition, because of the inevitable contact and uniformity issues of the PFETs, extraction methods based on multiple devices should be carefully applied for the extraction. However, by considering the parameters as gate bias-dependent power laws and dividing the drain current with transconductance ($I_{ds}/g_m$ method), current-voltage relations decoupled from the contact effect can be derived using a single transfer characteristic of the PFETs. Furthermore, the gate-dependent parameters, which are consistent with those obtained by the four-probe method, are successfully extracted using the relations. Moreover, using the proposed method, we can figure out that the enhanced device performances of the OTS-treated DPP-DTT PFETs are attributed to the minimized scattering and reduced contact resistance. Thus, we strongly believe that our proposed parameter extraction method based on a single device is a simple and reliable method for an in-depth study of device performance.

**Supplementary Materials:** The following supporting information can be downloaded at: https://www.mdpi.com/article/10.3390/cryst13071075/s1, Figure S1: Device parameter analyses of the HMDS-treated DPP-DTT PFETs using the *Y-function method*; Figure S2: Contact angle images of droplets DI water on the substrates; Figure S3: Device parameter analyses of the OTS-treated DPP-DTT PFETs.

**Funding:** The authors acknowledge the financial support from the Basic Science Research Program (Sejong Fellowship) through the National Research Foundation of Korea (NRF) funded by the Ministry of Education (NRF-2022R1C1C2008865).

**Institutional Review Board Statement:** Not applicable.

**Informed Consent Statement:** Not applicable.

**Data Availability Statement:** Not applicable.

**Conflicts of Interest:** The author declares no conflict of interest.

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
