# Peer review of "Analyzing Transfer Characteristics of Disordered Polymer Field-Effect Transistors for Intrinsic Device Parameter Extraction"

_crystals, doi:10.3390/cryst13071075_

Round 1

Reviewer 1 Report

The paper describes a method to determine the intrinsic mobility and contact resistance without the need of specific experimental geometries. It achieves this by reworking of the basic equations for the mobility, contact resistance and transconductance. The result is a set of equations that can be used on a single set of transfer characteristics without the need of special geometries.

The main area of uncertainty therefore comes in the derivation of the equations. First of a minor point in that some of the parameters are incorrectly labelled, I_d instead of I_ds and V_g instead of V_gs. However, more important and a point that needs clarification, is the justification for the assumptions made in equation 9.

When using the derived equations to obtain the mobility and contact resistance for PBTTT there is a factor two difference for both and while a better agreement is reached for the DPP-DTT, there is no discussion as what causes these differences.

On a side note, the font inside the figures is too small.

There are some minor modifications needed where the English is not correct. An example of this is the sentence on line 35-36.

Author Response

Reviwer1

Comments to the Author

The paper describes a method to determine the intrinsic mobility and contact resistance without the need of specific experimental geometries. It achieves this by reworking of the basic equations for the mobility, contact resistance and transconductance. The result is a set of equations that can be used on a single set of transfer characteristics without the need of special geometries.

  1. The main area of uncertainty therefore comes in the derivation of the equations. First of a minor point in that some of the parameters are incorrectly labelled, Id instead of Ids and Vg instead of Vgs. However, more important and a point that needs clarification, is the justification for the assumptions made in equation 9.

Response) Thank you for your valuable comments. We now correct the labels using Ids, Vds, and Vgs as below. Please check the revised manuscript.  

For the issue of the assumption in equation 9, when we compare the intrinsic mobilities from the 4-probe method to the extrinsic mobilities of PFETs, those of the HMDS-treated PBTTT TFTs were 0.0012 and 0.0013 cm2 V1 s1 at Vgs-Vth= 1 V. However, as the gate bias increases, the difference increases and is maximized at the gate bias of – 18 V, of which those were extracted to be 0.0092 and 0.0100 cm2 V1 s1, respectively. If the decreases in mobility due to the scattering are minimized, the difference would increase further as the gate bias increase. Thus, we regard the difference between the intrinsic and extrinsic mobilities at the Vgs-Vth= 1 V as negligible. To help readers understand this issue, we add the following sentences in the manuscripts. Thank you very much.

Please note that the intrinsic mobility from the 4-probe method to the extrinsic mobility of the HMDS-treated PBTTT TFTs (at Vgs-Vth= 1 V) was almost the same as extracted to be 0.0012 and 0.0013 cm2 V1 s1, respectively. Thus, we tried to derive the relations based on the assumption in Equations (9)(μint,0μext,0).

  1. When using the derived equations to obtain the mobility and contact resistance for PBTTT there is a difference for both and while a better agreement is reached for the DPP-DTT, there is no discussion as what causes these differences.

Response) We posit that the differences between the parameters by the 4-probe method and those by the proposed method (Ids/gm) are attributed to the increasing charge scattering in the channel as the gate bias increases. When the charge scattering happens severely, the discrepancy between the observed mobility and the power-law modelled field-effect mobility (μint=μint,0(Vgs-Vth)α) would increase, resulting the underestimation of the coefficient of α. As a result, the deduced contact resistance and mobility could be slightly under and overestimated than those by the 4-probe method. In this context, it is believed that the channel scattering effect in PBTTT PFETs is greater than those in D-A type DPP-DTT PFETs. As a result, the deduced parameters pf PBTTT could be slighlt more deviated than those of DPP-DTT. However, as in Table 1, the estimated contact resistances by both methods are in an order, and the differences in the mobilities are in the suborder of 10-3 cm2V−1s-1. Thus, we regard the proposed extraction method as reliable. To help readers understand this issue, we add the following sentences in the manuscripts. Thank you very much.

In addition, probably due to the charge scattering, the deduced contact resistance and mobility by the Ids/gm method are slightly under and overestimated than those by the 4-probe method. As the gate bias increases, the discrepancy between the observed mobility and the power-law modeled field-effect mobility (μint=μint,0(Vgs-Vth)α) would increase and result in the underestimation of the coefficient of α. As a result, the deduced contact resistance and mobility could be slightly under and overestimated than those by the 4-probe method. For PBTTT PFETs, the channel scattering effect could be greater than those in D–A type semiconducting copolymers of DPP-DTT, resulting in more deviated parameters. However, the estimated contact resistances by both methods are in an order, and the differences in the mobilities are in the suborder of 10-3 cm2V−1s-1. Thus, we regard the parameter extraction method of the Ids/gm method as reliable.

  1. On a side note, the font inside the figures is too small.

Response) We now increase the font size of all the figures for better visibility as below. Please check the revised manuscript. Thank you very much.

Reviewer 2 Report

Dear Editor,

I reviewed the manuscript “Analyzing transfer characteristics of disordered polymer field-effect transistors for intrinsic device parameter extraction” by Minho Yoon, where an intrinsic device parameter extraction method based on a single device has been investigated for disordered PFETs. The work is clear and could be helpful to shed light on the phenomena based on devices performance, although I have some major comments that must be take into account:

-        The authors claimed that 300 nm-thick p+-Si/SiO2 wafer was used as the substrate. I think this requires more information must be added, since this kind of material can adopt different three-dimensional dispositions (i.e. polymorphs). The interface phenomena are in fact strictly correlated to this aspect, also able to modify the performance of the device.

-        Lines 86 to 103 of page 3, including equations 2 and 3 should be reported in materials and methods section, and not in results one, since they explain the methods and the approaches. 

-        In the conclusions section, the authors simply repeat what they reported in the abstract and in the introduction, without significative differences. I suggest to rewrite this section emphasizing the innovative and characteristic aspects of their work.

Author Response

Summary of Revisions & Responses to Reviewers' Comments

We sincerely appreciate the efforts of the reviewers, who provided important and helpful comments on the manuscript. We have revised the manuscript to address the reviewers’ comments. Our response to each comment is given below in detail. In the following, the reviewers’ comments are in black fonts and our replies are in blue fonts.

Reviwer 2

I reviewed the manuscript “Analyzing transfer characteristics of disordered polymer field-effect transistors for intrinsic device parameter extraction” by Minho Yoon, where an intrinsic device parameter extraction method based on a single device has been investigated for disordered PFETs. The work is clear and could be helpful to shed light on the phenomena based on devices performance, although I have some major comments that must be take into account:

  1. The authors claimed that 300 nm-thick p+-Si/SiO2 wafer was used as the substrate. I think this requires more information must be added, since this kind of material can adopt different three-dimensional dispositions (i.e. polymorphs). The interface phenomena are in fact strictly correlated to this aspect, also able to modify the performance of the device.

Response) We have used the widely used and commercially available 300 nm-thick p+-Si/SiO2 wafer. Although the orientation of the Si is (100), the thermally grown 300 nm thick SiO2 is amorphous, not crystalline as quartz. The wafers are easy to use, but the surface contains hydroxyl groups, which act as trap sites. Thus, as described in the manuscript, we treated the surface of the SiO2 with HMDS for minimizing the interface traps. If we use the crystalline SiO2 as dielectrics by other deposition methods, the interface phenomena should be investigated in more detail. However, we believe this could be beyond this study. Thank you very much.  

  1. Lines 86 to 103 of page 3, including equations 2 and 3 should be reported in materials and methods section, and not in results one, since they explain the methods and the approaches.

Response) As you pointed out, we added the sentences and equations in the material and methods section, and modify the manuscript as below. Thank you very much.

Figure 1(b) displays the representative transfer characteristics (Ids vs. Vgs) of the PBTTT-based PFETs; it exhibits p-type characteristics, of which the field-effect mobility (μext) from the transconductance was estimated to be ~ 0.009 cm2 V−1 s−1 along with a high on/off ratio (>103). The gate leakage current level was maintained as low as 10-10 A. Figure 1(c) presents the corresponding output characteristics (Ids vs. Vds) of the PFETs. Then, for extracting the referential intrinsic parameters of the PFETs, we investigated the transfer characteristics with the four-probe method. Please see the material and methods section for more information on the four-probe method. Figure 2(a) shows the measured channel and contact potentials as a function of the gate bias at the drain bias of –1, – 3, and – 5 V.

  1. In the conclusions section, the authors simply repeat what they reported in the abstract and in the introduction, without significative differences. I suggest to rewrite this section emphasizing the innovative and characteristic aspects of their work.

Response) Thank you for your valuable comments. We rewtire the conclusion section for emphasizing the significance of this work as below. Thank you very much.

In this study, we investigate an intrinsic device parameter extraction method for disordered polymer field-effect transistors (PFETs). Due to the localized states of the disordered polymer semiconductors, the field-effect mobility and contact resistance are strongly dependent on the gate bias. In addition, because of the inevitable contact and uniformity issues of the PFETs, extraction methods based on multiple devices should be carefully applied for the extraction. However, by considering the parameters as gate bias-dependent power laws and dividing the drain current with transconductance (Ids/gm method), current-voltage relations decoupled from the contact effect can be derived using a single transfer characteristic of the PFETs. Furthermore, the gate-dependent parameters, consistent with those obtained by the four-probe method, are successfully extracted using the relations. Moreover, using the proposed method, we can figure out that the enhanced device performances of the OTS-treated DPP-DTT PFETs are attributed to the minimized scattering and reduced contact resistance. Thus, we strongly believe that our proposed parameter extraction method based on a single device is a simple and reliable method for an in-depth study of device performance. 

Round 2

Reviewer 2 Report

The Authors answered to all questions raised.